# The Association of Gender and Body Mass Index on the Values of Static and Dynamic Balance of University Students (A Cross-Sectional Design Study)

**George Danut Mocanu [1] and Gabriel Murariu [2,*]**

1    Faculty of Physical Education and Sport, "Dunărea de Jos" University of Galati, 800008 Galati, Romania; george.mocanu@ugal.ro
2    Faculty of Sciences and Environment, "Dunărea de Jos" University of Galati, 800008 Galati, Romania
*    Correspondence: gabriel.murariu@ugal.ro; Tel.: +40-74-012-6940

**Abstract:** The balance of the body conditions the quality and efficiency of the movements in daily and sports activities, its impairment generating problems in the manifestation of motor skills for all age groups. The aim of this study is to analyze how the gender and BMI values of university students influence the results of the static and dynamic balance tests applied. The investigated group consists of 195 undergraduate students, from various specializations of the Dunărea de Jos University in Galați (99 males and 96 females, ages = 20.16 ± 1.98, BMI = 24.15 ± 5.68). The independent variables gender and BMI levels (underweight, normal weight and overweight/obese) were defined. The participants were evaluated in May 2019 using a series of 7 tests: one leg standing test with eyes closed, stork test, flamingo test, Bass test, functional reach test, walk and turn field sobriety test and Fukuda test. The results provided using multivariate analysis (MANOVA) indicate balanced performance for the gender variable, but with significantly better values for women in the one leg standing test, flamingo test and functional reach test (F values are associated with thresholds $p < 0.05$). Men obtain slightly better, but statistically insignificant ($p > 0.05$), performances on the stork test, Bass test and Fukuda test. The BMI step comparison confirms the difficulties of the overweight group in assessing balance, with the lowest scores in maintaining static positions and the most errors in dynamic balance tests, with significant differences from normal and underweight in most tests ($p < 0.05$). An interesting aspect is the slightly superior performance of the underweight compared to the normal weight group, for the one leg standing test, flamingo test and walk and turn field sobriety test. The analysis of Spearman correlation coefficients indicates a number of significant associations between elevated BMI values and decreased performance on balance tests.

**Keywords:** balance; students; stability; gender; obesity

## 1. Introduction

Even though it is not included in the main components of physical fitness, balance training plays a major role in improving body stability and performance training, especially for the obese, whose risk of falling and injury is high, according to [1]. Static standing balance is a major component of motor fitness. There are studies that highlight the importance of mid-lateral postural control for good balance and movement efficiency in dynamic motor activities [2]. Balance tests applied to Algerian students (x = 13 years) assessed the postural control deficit, caused by muscle fatigue, which decreases performance in motor activities, so they must be introduced as elements of the FitnessGram test series. This aspect supports the importance of balance as a core element of functional fitness, with links to torso muscle strength, endurance and flexibility [3].

The types of daily activity require demands on balance. The ability of people to maintain balance and to shift weight when driving an SPMD/self-balancing personal mobility device (such as a bicycle, scooter, electric moped) is affected by target distance

and target direction, during upright standing [4]. Young people's interest in spending time in various outdoor activities is high should not be affected by modern technologies (mobile telephony) or lost time with transport [5]. For Kosovo teenagers (13–15 years old group) there is a better involvement of boys in moderate to vigorous physical activity, and 13-year-old girls are more involved if they are supported by their teacher and colleagues [6]. The situation is similar for normal weight teenage boys in Kosovo (15–18 years old) who have the support of the teacher and colleagues and are more physically active than girls of the same age [7].

The effects of mobile phone use on posture during static and dynamic balance assessment tasks were investigated in a group of Taiwanese students (x = 21.75 years). Sending messages affects postural stability. The task of sending the message is also perceived to be more difficult, but younger subjects have superior adaptability and manage to maintain balance, easily preventing falls [8]. The negative effect of wearing school bags for school children (11–15 years old) is highlighted by the use of a baropodographic platform. The excess weight loads the areas of the heels and metatarsals, causing their flattening and affecting balance, through the incorrect posture of the body [9].

Obesity is a problem of the younger generations in European countries. These students usually have a low involvement in school and free time physical activities, thus needing curricular changes in physical education in terms of curriculum content and teaching methods [10]. Involvement in physical activity (PA) for students in physical education faculties in Europe differs by gender. With the exception of Germany, for the rest of the countries analyzed (Poland, the Netherlands and the Czech Republic) there are higher levels of PA for men [11]. Obese Japanese girls have lower scores on static and dynamic balance tests than adult men, so they have a higher risk of falling and injuring themselves during physical activity, according to [12]. In older adults, negative associations are found between BMI values and performance on balance tests, however, physically active overweight people have higher balance values, and thus, the amount of physical activity favorably influences postural control. Physically inactive people have poorer balance scores, and improving muscle strength through physical activity is a factor that improves postural stability. [13]. Poor balance is associated with the risk of ligament injury. Its improvement reduces these problems, and the optimization of coordination processes also favorably influences the performance of balance tests for adolescent soccer players [14]. Balance is defining for correct technical executions in sports (soccer), having a role in injury prevention, and exercises for its development are developing sensory and motor skills that ensure performance [15]. The positive influence of football training on the postural stability and balance of young women is clear. Female participants show a better use of the vestibular system and the asymmetrical support of body weight, compared to sedentary women [16]. The study of the balance in the pubertal stage and its relationship with the dominance of the cerebral hemispheres is defining for the optimization of the selection in technical tests in athletics [17,18]. Strong positive correlations between the performance of snipers in India and the results of the stork static balance test are identified by [19].

The manifestation at a higher level of static and dynamic balance is conditioned by the state of the visual, proprioceptive and vestibular systems [20]. The low level of maturity of these systems generates postural disorders for teenagers (13–21 years) with intellectual disabilities (IDs), especially for boys with moderate disability ID, compared to those with mild disability ID, both in maintaining open-eyed positions as well as closed [21]. The existence of visual biofeedback significantly improves balance by testing it on a computerized wobble board, according to [22]. Workouts based on sit-to-stand physical exercises, combined with real-time visual feedback, will significantly improve the parameters related to the strength of the leg muscles, the quality of walking and the balance of the body, according to [23]. However, young people with visual and hearing disabilities can achieve well-being by forming daily patterns of mobility around the house and at school, thus creating pathways to resilience [24]. The role of the visual analyzer in the postural stability of young ballet dancers is very important. The 18-year-olds group have better values than

14-year-olds, with poor results for static balance with their eyes closed [25]. Proprioceptive information at the ankle, along with other sensory information, has a defining role in postural control, maintaining balance in sports activities and achieving performance in competitions. A higher balance reduces the chances of lower limb injuries [26]. Cervical proprioception provides additional information in addition to that provided by vestibular and auditory analyzers. A recent study found associations between kinesthetic information in the neck and balance on the non-dominant/non-preferred leg of Taekwondo fighters, as an adaptation that is not present in the untrained [27].

The physical fragility of the obese elderly is obvious, associated with the impairment of the sensory component, poor muscle quality, limited physical training, lack of strength, reduced walking speed and greater body balance in orthostatism, and the amount of PA/physical activity influences postural control [13,28,29]. Significant differences are found between obese and normal weight, related to CoP/center of pressure and velocity and displacement in the antero-posterior and mid-lateral plane. For obese women there are significant differences only in the antero-posterior plane, but for men there are also differences in the mid-lateral plane/ML displacements [30].

A comparison of static and dynamic athletic performance related to balance was carried out for those engaged in sports games in Iran. A significant difference for static balance (stork test) is between the volleyball and handball groups, in favor of volleyball. For dynamic balance (YBT/Y balance test) significant differences are reported between the handball and volleyball teams, respectively, basketball and volleyball, so the specificity of the effort influences performance in balance tests [31]. For static and dynamic balancing tests, performed before and after a dance session (Hungarian folk dancers), via examination with eyes closed and open, greater deviations of the joints are found on the non-dominant leg, and oscillations in the posterior direction are significantly higher after dancing [32].

The increased intensity of physical activities generates a reduction in the values of static balance after effort (One leg standing test), but a break of 15 min will generate an increase in the values for this test [33]. The need for individualized exercise programs for obese children, aimed at increasing mobility and stability of the ankle, improving postural balance, stimulating the proprioceptive system and strengthening the lower limb muscle groups are stipulated by [34]. They recommend static and dynamic exercises, with eyes closed and open, performed on various surfaces (stable and unstable). The idea of the efficiency of unstable surfaces is also supported by [35]. A comparative study of volleyball players (elite level, x = 23.6 years) shows that the application of classic balance training for 12 weeks is more effective than the variant of sand training, which leads to worsening of static balance (evaluated using the Romberg test), in the opinion of [36].

Applying yoga exercises/asanas to obese young people (21–25 years old) produced significant improvements for the static and dynamic balance evaluated using one leg standing balance and a functional reach test [37]. A similar study at the level of students (18–25 years old) confirms the effectiveness of yoga asanas for the performance of static balance (stork test) and muscle strength, according to [38]. Applying balance training for obese people (20–50 years old) for 4 weeks improved their static and dynamic performance [39].

Implementing a dynamic core exercise (DCE) program in the warm-up part of physical education lessons for 6 weeks for 10–11 year olds has beneficial and significant effects on physical parameters., flexibility and balance, according to [40]. A solution to improve the balance for those sitting on a chair/sitting postural balance for about 10 h a day is based on the use of a linear actuator and MR damper. The device offers combined exercises to strengthen the muscles of the back that are too weak and to relax the contracted one, regulating the muscular activity of the trunk between the dominant and the non-dominant part [41].

Applying an artistic gymnastics program (12 weeks × 2 workouts/week × 60 min/session) generated improvements in results related to static balance (flamingo test), strength and flexibility for students in physical education and sports in Turkey [42]. A 6-week

program based on Wii Fit exergames had positive effects on static balance and motor competence for young normal weight and obese college students [43].

*The purpose of the study* is to investigate the way in which the gender and the classification of students in different BMI levels influence the applied performances in static and dynamic balance tests, with identification of the registered differences and comparison of the results obtained in other scientific studies.

*Working hypotheses*:

**H1.** *We estimate that the independent gender variable will generate significant differences in the registered dependent variables (values obtained from the applied equilibrium tests).*

**H2.** *We estimate that the independent variable BMI steps (with the underweight, normal weight and overweight steps) will generate significant differences between the performances obtained, for the entire set of tests used in the assessment of balance.*

**H3.** *We assume that we will obtain strong and significant correlations between BMI values and performance on balance tests, in the sense that poor performance on tests will be associated with an increase in the BMI.*

## 2. Materials and Methods

### 2.1. Participants

The investigated group initially consisted of 216 undergraduate students from the Dunărea de Jos University in Galați, with various specializations: Pharmacy, Dentistry, Computers and Information Technology, Automation and Applied Informatics, Applied Electronics, Electrical Engineering. Out of these, 7 students who were engaged in performance sports were excluded from the study (so that this factor did not affect the results, most of those investigated not having constant concerns about physical activities, apart from physical education lessons) and 14 students who did not participate in the entire set of tests. In total, 195 students remained in the study (ages = 20.16 ± 1.98, weight = 72.70 ± 21.37, height = 172.53 ± 10.16, BMI = 24.15 ± 5.68). The studied group was formed via random selection of student groups/classes from several faculties, following and ensuring gender balance (99 men and 96 women). The calculation of individual BMI values was done using the formula weight (kg)/[height (m)]$^2$. However, following the processing of personal data parameters and the classification by BMI levels, we found an obvious imbalance between the percentages of each BMI stage, with very low values for the underweight category, at the level of the whole group and by gender. These data are summarized in Table 1. We mention that the obese participants (with BMI > 30, 37 as number, which represents 18.97% of the total investigated group) were included in the same group with the overweight ones, in order to facilitate the statistical calculation, by reducing the number of compared data pairs. All participants had a favorable medical opinion for engaging in physical activity.

**Table 1.** Distribution of participants and related percentages by gender and BMI steps.

| Gender | Participants | Underweight BMI (Below 18.5) | Normal Weight BMI (18.5–24.9) | Overweight/Obesity BMI (25–30)/BMI (above30) |
|---|---|---|---|---|
| **Male** | 99 (50.76%) | 6 (3.07%) | 51 (26.15%) | 42 (21.54%) |
| **Female** | 96 (49.23%) | 15 (7.69%) | 60 (30.76%) | 21 (10.76%) |
| **Total lot** | 195 (100%) | 21 (10.77%) | 111 (56.92%) | 63 (32.30%) |

### 2.2. Procedure

The applied work methodology is the one specific to cross-sectional research. The investigation and data collection presented in this study were conducted at the end of the 2018–2019 academic year (May), before the COVID 19 pandemic, at the Research Center

for Human Performance, belonging to the Faculty of Physical Education and Sports in Galați. A total of 7 balance tests were applied, of which 3 involved static balance assessment (one leg standing with eyes closed/sec, stork test/sec and flamingo test/falls) and 4 were dynamic balance investigation tests (Bass test/points, functional reach test/cm, walk and turn field sobriety test/errors and Fukuda test/degrees of rotation); description, scores, validity and scientific reliability were available through sources [44–52]. It should be noted that the Fukuda, flamingo and walk and turn field sobriety tests are atypical, i.e., higher scores actually indicate poorer performance, which is important in analyzing the signs of correlation coefficients. It was preferred to test the subjects in small working subgroups (5–7 people), to ensure peace and focus. The execution technique was explained to the students, including the requirements and mistakes specific to each test, and they were allowed to become familiar with their specifics. They were advised not to exert intense effort before the test, in order to avoid the onset of muscle and mental fatigue, which would have affected the performance of the balance tests. The warm-up was based on light aerobic effort and dynamic stretching movements, the latter are also recommended during the test breaks, in order to maintain an optimal muscle tone.

The circadian rhythm seems to affect balance performance, most sources indicating weaker values in the early morning and late evening, with better values in the afternoon [53–55]. Exceptions are elite athletes, where these fluctuations are not observed, but they are present in untrained teenagers, according to [56]. For this reason, we chose the 12–16 p.m. interval for testing student groups.

All participants were instructed regarding the purpose of the study, respecting the rules of ethics of scientific research with human subjects, related to the protection and processing of personal data, in accordance with the Helsinki Declaration [57,58].

### 2.3. The Statistical Analysis of Data

The statistical calculation was based on the use of SPSS software (Verse 24). Anova parametric techniques (multivariate and univariate test) were applied with the determination of F values, significance thresholds and partial eta squared/$\eta^2_p$, Levene's test of equality of error variances, analysis of the significance of differences in mean values per pair for gender variables and BMI steps (with the application of Bonferroni post hoc tests) [59–61]. As the normal data distribution curve for most tests is not observed, the non-parametric variant (Spearman) was used to calculate the correlation coefficients between the BMI values and the results of the balance tests. Comparative representation graphs for the average results between the genres for each BMI step were created using Microsoft Excel editor. The confidence interval was set at 95% ($p < 0.05$) [62–64].

### 3. Results

The results obtained from the statistical data processing are represented in tables and graphs: Table 2 (multivariate analysis), Tables 3 and 4 (univariate analysis, with the identification of the effect of each independent variable on test results), Tables 5 and 6 (analysis on data pairs of mean values and the significance of differences between them), Table 7 (correlations between BMI values and results on balance tests), graphs 1–3 (comparison by gender of average performance at each BMI stage).

**Table 2.** The results of the multivariate tests [a] (MANOVA).

| Effect | λ | F | Hypothesis df | Error df | Sig. | $\eta_p$ | Observed Power |
|---|---|---|---|---|---|---|---|
| Gender | 0.675 | 12.572 [b] | 7.000 | 183.000 | 0.000 | 0.325 | 1.000 |
| BMI framing steps | 0.686 | 5.425 [b] | 14.000 | 366.000 | 0.000 | 0.172 | 1.000 |
| Gender * BMI framing steps | 0.850 | 2.219 [b] | 14.000 | 366.000 | 0.007 | 0.078 | 0.970 |

[a] Design: Gender + BMI framing steps + Gender * BMI framing steps. [b] Exact statistic. λ—Wilk's lambda; F—Fisher test; df—degrees of freedom; Sig.—level of probability; $\eta^2_p$—partial eta squared.

**Table 3.** Univariate test results (ANOVA)—the influence of the gender variable on performance in the balance tests.

| Dependent Variable | Sum of Squares | Mean Square | F (1, 193) | Sig. | Partial Eta Squared | Noncent Parameter | Observed Power |
|---|---|---|---|---|---|---|---|
| One standing balance test | 498.941 | 498.941 | 14.185 | 0.000 | 0.068 | 14.185 | 0.963 |
| Functional reach test | 157.890 | 157.890 | 4.017 | 0.046 | 0.020 | 4.017 | 0.514 |
| Stork test | 30.826 | 30.826 | 2.738 | 0.100 | 0.014 | 2.738 | 0.377 |
| Flamingo test | 659.669 | 659.669 | 43.920 | 0.000 | 0.185 | 43.920 | 1.000 |
| Bass test | 412.554 | 412.554 | 2.164 | 0.143 | 0.011 | 2.164 | 0.310 |
| Walk and turn field sobriety test | 0.640 | 0.640 | 1.979 | 0.161 | 0.010 | 1.979 | 0.288 |
| Fukuda test | 0.344 | 0.344 | 0.000 | 0.982 | 0.000 | 0.000 | 0.050 |

**Table 4.** Univariate test results (ANOVA)—influence/effect of the variable BMI framing steps on performance on balance tests.

| Dependent Variable | Sum of Squares | Mean Square | F (2, 192) | Sig. | Partial Eta Squared | Noncent Parameter | Observed Power |
|---|---|---|---|---|---|---|---|
| One Standing balance test | 617.602 | 308.801 | 8.889 | 0.000 | 0.085 | 17.779 | 0.971 |
| Functional reach test | 33.781 | 16.891 | 0.421 | 0.657 | 0.004 | 0.841 | 0.118 |
| Stork test | 108.109 | 54.055 | 4.953 | 0.008 | 0.049 | 9.906 | 0.805 |
| Flamingo test | 975.599 | 487.800 | 36.261 | 0.000 | 0.274 | 72.522 | 1.000 |
| Bass test | 5455.519 | 2727.760 | 16.499 | 0.000 | 0.147 | 32.998 | 1.000 |
| Walk and turn field sobriety test | 8.537 | 4.268 | 15.034 | 0.000 | 0.135 | 30.068 | 0.999 |
| Fukuda test | 5778.505 | 2889.252 | 4.371 | 0.014 | 0.044 | 8.741 | 0.751 |

**Table 5.** Analysis of the significance of the differences between the average values in the gender balance tests (male = 99, female = 96).

| Test/Dependent Variables | Group | Mean | Std. Deviation | Std. Error | a-b | Sig. [b] |
|---|---|---|---|---|---|---|
| One standing balance test | Male | 5.148 | 3.457 | 0.596 | −3.200 * | 0.000 |
| | Female | 8.348 | 7.689 | 0.605 | | |
| Functional reach test | Male | 40.762 | 6.957 | 0.630 | −1.800 * | 0.046 |
| | Female | 42.562 | 5.469 | 0.640 | | |
| Stork test | Male | 4.334 | 4.008 | 0.337 | 0.795 | 0.100 |
| | Female | 3.538 | 2.508 | 0.342 | | |
| Flamingo test | Male | 8.272 | 4.597 | 0.390 | 3.679 * | 0.000 |
| | Female | 4.593 | 2.950 | 0.396 | | |
| Bass test | Male | 70.565 | 12.900 | 1.388 | 2.909 | 0.143 |
| | Female | 67.656 | 14.681 | 1.409 | | |
| Walk and turn field sobriety test | Male | 0.333 | 0.589 | 0.057 | 0.115 | 0.161 |
| | Female | 0.218 | 0.546 | 0.058 | | |
| Fukuda test | Male | 22.353 | 25.112 | 2.635 | −0.084 | 0.982 |
| | Female | 22.437 | 27.317 | 2.676 | | |

* The mean difference is significant at the 0.05 level. [b] Adjustment for multiple comparisons: Bonferroni.

**Table 6.** Analysis of the significance of the differences between the mean values on BMI steps in the balance tests.

| Test | Group | Mean | Std. Deviation | Std. Error | a-b | Sig.[b] | a-c | Sig.[b] | b-c | Sig.[b] |
|---|---|---|---|---|---|---|---|---|---|---|
| One standing Balance test | a. underweight | 11.557 | 9.936 | 1.286 | | | | | | |
| | b. normal weight | 6.610 | 6.135 | 0.559 | 4.947 * | 0.002 | 6.245 * | 0.000 | 1.298 | 0.492 |
| | c. overweight | 5.312 | 2.990 | 0.743 | | | | | | |
| Functional reach test | a. underweight | 41.857 | 4.855 | 1.383 | | | | | | |
| | b. normal weight | 41.950 | 5.630 | 0.601 | −0.093 | 1.000 | 0.810 | 1.000 | 0.903 | 1.000 |
| | c. overweight | 41.047 | 7.778 | 0.798 | | | | | | |
| Stork test | a. underweight | 4.213 | 4.347 | 0.721 | | | | | | |
| | b. normal weight | 4.498 | 3.838 | 0.314 | −0.285 | 1.000 | 1.342 | 0.326 | 1.627 * | 0.006 |
| | c. overweight | 2.871 | 1.245 | 0.416 | | | | | | |
| Flamingo test | a. underweight | 4.047 | 3.556 | 0.800 | | | | | | |
| | b. normal weight | 5.099 | 2.954 | 0.348 | −1.051 | 0.689 | −5.619 * | 0.000 | −4.568 * | 0.000 |
| | c. overweight | 9.666 | 4.700 | 0.462 | | | | | | |
| Bass test | a. underweight | 66.000 | 14.679 | 2.806 | | | | | | |
| | b. normal weight | 73.639 | 12.584 | 1.220 | −7.640 * | 0.040 | 3.762 | 0.741 | 11.402 * | 0.000 |
| | c. overweight | 62.238 | 12.708 | 1.620 | | | | | | |
| Walk and turn field sobriety test | a. underweight | 0.000 | 0.000 | 0.116 | | | | | | |
| | b. normal weight | 0.162 | 0.437 | 0.051 | −0.162 | 0.607 | −0.571 * | 0.000 | −0.409 * | 0.000 |
| | c. overweight | 0.571 | 0.734 | 0.067 | | | | | | |
| Fukuda test | a. underweight | 17.190 | 21.671 | 5.611 | | | | | | |
| | b. normal weight | 18.927 | 23.760 | 2.440 | −1.737 | 1.000 | −13.048 | 0.136 | −11.310 * | 0.017 |
| | c. overweight | 30.238 | 29.899 | 3.239 | | | | | | |

* The mean difference is significant at the 0.05 level. [b] Adjustment for multiple comparisons: Bonferroni.

**Table 7.** The value of Spearman correlations between BMI values and performance in balance tests.

| | | BMI | One Leg Standing | Functional Reach | Stork | Flamingo | Bass | Walk and Turn | Fukuda |
|---|---|---|---|---|---|---|---|---|---|
| BMI | ρ (rho) | 1.000 | −0.106 | −0.067 | −0.197 * | 0.468 * | −0.212 ** | 0.414 * | 0.261 * |
| | Sig. | . | 0.139 | 0.352 | 0.006 | 0.000 | 0.003 | 0.000 | 0.000 |

* Correlation is significant at the 0.05 level (2-tailed). ** Correlation is significant at the 0.01 level (2-tailed).

The multivariate analysis indicates the overall effect of the 2 independent variables (gender and BMI framing steps) and the interaction between them on the whole set of applied tests, with statistically significant results ($p < 0.05$), in all 3 variants in Table 2. We notice that the strongest influence on performance of balance is the gender variable (32.5% of the variance in test results is explained by this variable) followed by BMI framing steps (17.2% of performance variation is due to BMI values). The gender * BMI framing steps interaction explains only 7.8% of the variance of the test results.

The univariate test analysis at the level of the independent variable gender (Table 3) indicates its effect, separately for each dependent variable (test applied at the level of the investigated group). Gender differences lead to significant thresholds and higher size effect scores (expressed via partial eta squared) for: one standing balance test ($p = 0.000$), with 6.8% of the performance variance due to gender, flamingo test ($p = 0.000$), with 18.5% of the variance of performance explained by the influence of gender, and the functional reach test ($p = 0.046$), with only 2% of the variance of results determined by the independent variable gender. For the other tests, no significant and strong effects of the gender variable on performance related to balance are noticed.

The univariate test analysis at the level of the independent variable BMI framing steps (Table 4) identifies significant effects for 6 out of the 7 tests to assess static and dynamic balance. The strongest influences are reported from high to low for: flamingo test ($p = 0.000$ and 27.4% of the performance variance explained by the BMI steps), Bass test ($p = 0.000$ and 14.7% of the variance), one standing balance test ($p = 0.000$ and 8.5% of the variance), walk and turn field sobriety test ($p = 0.000$ and 13.5% of the variance), stork test ($p = 0.000$ and 4.9% of the variance) and Fukuda test ($p = 0.012$ and 4.4% of the variance assigned to the independent variable). The only situation with insignificant influence is for the

functional reach test ($p = 0.657$ and only 0.4% of the variance explained by the influence of BMI framing steps).

The comparative analysis of the average values between pairs, at the level of the independent gender variable, identifies 3 cases of significant differences, all in favor of women: one standing balance test ($p = 0.000$), flamingo test ($p = 0.000$) and functional reach test ($p = 0.046$), according to Table 5. Thus, hypothesis 1 (H1) is partially confirmed. The group of women has better values (makes fewer mistakes) also in the walk and turn field sobriety test, but no significant differences were reported ($p = 0.161$). The superior result of the group of women in the functional reach test could be favorably influenced also by the better level of flexibility of the spine (specific to the female gender), so it must be viewed with caution. The male group has slightly higher scores in only 3 of the tests, but no significant differences were reported: stork test ($p = 0.100$), Bass test ($p = 0.161$) and Fukuda test ($p = 0.982$). It is difficult to argue that the men's group performs slightly better in the stork test than the women's group, while in the one standing balance test (which also assesses static balance, but with their eyes closed) the values are very poor. A possible explanation could be the better level of strength for the muscle groups that are supported by isometric contraction, maintaining a balanced position on the tip of the supporting leg.

The evaluation of the 2 groups identified major problems for taking the flamingo test in men. A total of 16 subjects needed 15 attempts (representing 15.5% of all men) to achieve 60 s of maintaining balance on the wooden beam, this being the maximum number of attempts/falls allowed by the test application methodology, and 9 men needed 14 attempts/falls, aspects that were not encountered in the case of women. Only 25 students in the whole group (representing 12.82%) performed the test with a maximum score/single attempt/fall that totaled 60 sec, of which 22 were women (representing 11.28% of the whole group) and only 3 men (representing 1.54% of the whole group), aspects that explain the very big differences between the genders for this test.

Regarding the Bass test, we noticed problems especially when trying to maintain balance on the markings/chips stuck on the ground (for 5 s), from one jump to another. Most of the students, however, made quite precise jumps on the markings, managing to cover these marks with the front of the foot/forefoot, when landing with the change of the support foot on the ground. Although the Bass test is classified as a variant of assessing dynamic balance, the holding phases on the landing leg are the ones that raised problems for both genders, and for this reason we consider that the test partially reflects the level of manifestation of unipodal static balance. The maximum score of 100 points is reached only by 3 participants (all men), which represents 1.53% of the total number of students investigated. Very low scores (below 50 points) are obtained by 12 students (9 women and 3 men, representing 6.15% of the tested group), with a poor spatial orientation and a limited accuracy of jumping on the markings on the ground.

Along with the Bass test, the Fukuda test is certainly the one that aroused major interest from those evaluated, due to its atypical nature and novelty, with all participants in the study performing it for the first time. It should be noted that most of those tested failed to maintain the position/starting point, with frequent movements in the forward direction, from a few cm to values of 1–2 m, even if these aspects are not quantified in the test. The average performance values of both sexes are balanced and below the threshold of 30 degrees, a benchmark that indicates the manifestation of vestibular disorders on the side of body rotation, but there are individual cases with extreme scores, reaching even over 90 degrees. The rotation of the body to the right side is dominant within the group, for 126 participants (representing 64.61% of the total students), the rotation on the left side is reported for 48 cases (representing 24.61% of the total students) and only 21 students (10.76% of the total) have no deviations/rotation of the body, after completing the 50 steps with lifting the knees up and eyes closed.

The averages for better performance belong to the groups of normal weight and underweight, compared to overweight (who have the worst average results), an aspect confirmed for all tests, according to the data presented in Table 6. Comparison of average

values by BMI classes brings most significant differences at the level of the 3 defined subgroups. With the exception of the functional reach test, where the slightly better results of the underweight and normal weight are not statistically significant compared to the overweight ones, for the other tests significant differences are reported for at least one of the analyzed pairs, so Hypothesis 2 (H2) is partially confirmed.

The very good result of the group of underweight in the one standing balance test and the significant differences resulting from the comparison with the average values of normal and overweight ($p = 0.000$) in both cases represent a surprise. After a careful analysis of Table 1 we notice that the underweight group consists mostly of women (15 students and only 6 men), the better values of women (previously analyzed in Table 5) generating this spectacular difference. However, this test does not show significant differences between normal and overweight ($p = 0.492$).

The best average results of the stork test belong to normal weight students, but the difference compared to the value of the underweight group is not significant ($p = 1.000$), but it is significant compared to overweight ($p = 0.006$). However, no significant differences are reported between the average values of underweight and overweight ($p = 0.326$). For the flamingo test, the best average results belong to the underweight and then the normal weight group, but with insignificant differences between these groups ($p = 0.689$). Poor overweight results generate significant differences compared to the average values of the other 2 groups ($p = 0.000.$). Of the 16 men who needed 15 attempts (threshold) to complete the test, none is normal weight; only 1 is underweight, with the other 15 falling into the class of overweight and obese. For the Bass test, there is a big difference between the values of normal weight compared to those of underweight and overweight ($p = 0.040$, respectively $p = 0.000$), but there is also lack of significance between the average values of underweight and overweight ($p = 0.741$).

The walk and turn field sobriety test demonstrates the problems related to the dynamic balance of the body while walking, for the category of overweight, which have the most errors associated with the test, the most common being related to the difficulty of returning to $180^0$ and exceeding the line marked on the ground. Rarely did mistakes occur in those in the normal weight group, and those in the underweight group followed the respective route without making mistakes. The differences are significant for the comparison between underweight and overweight, respectively between normal and overweight ($p = 0.000$), and are insignificant between the group of normal and underweight ($p = 0.607$).

The Fukuda test identifies the vestibular problems of the 3 groups, expressed in degrees of rotation on one side. Overweight people have the largest deviations, with average values exceeding the threshold of 300. However, the high average values of this group are due to the extreme scores recorded by some students; of the 9 cases with values $\geq 900$, 6 are overweight and only 3 normal weight. Normal weight students have the best results in this test, followed by underweight, but the only significant difference is between normal weight and overweight ($p = 0.017$).

The analysis of the associations between BMI values and the results of the balance tests in Table 7 indicates a decrease in performance when BMI values increase and negative correlations for 4 tests, without significance only for the one leg standing test and the stork test. The positive correlations reported in the flamingo, walk and turn and Fukuda tests indicate an increase in the number of falls, errors or degrees, so also an undesirable influence of high BMI values. Although the values of the Spearman coefficient indicate weak and moderate correlations, the fact that they are statistically significant partially confirms the third working hypothesis (H3).

The comparative average values (by gender) of the balance tests for the group of underweight are graphically represented in Figure 1. They cannot be generalized, due to the fact that the number of underweight subjects is the lowest in our group, so the resulting data must be analyzed with reservation. The very high average of women underweight for the one leg standing test influences the average of the whole group of women, thus explaining the gender superiority for this test. Underweight men get better

average performances for the stork test and Bass test, but also in the case of the functional reach test (except for the average values between genders for the whole group, where women have higher scores). For the Fukuda test, the much lower values of underweight women are noticeable, and both genders walked the route for the walk and turn field sobriety test without errors. A different situation is presented for normal weight male and female case (Figure 2).

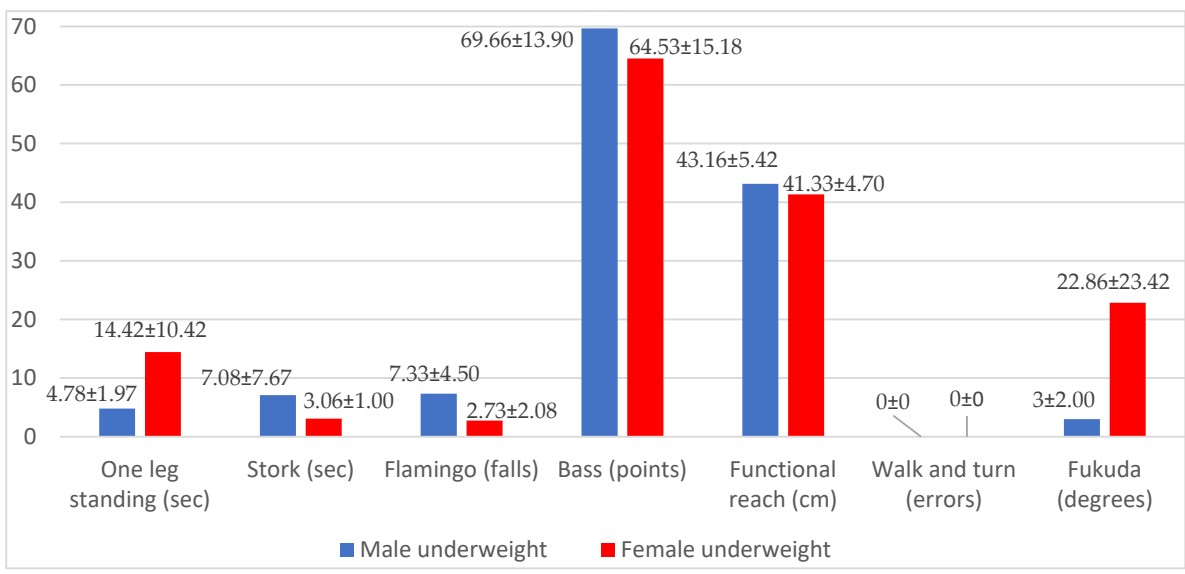

**Figure 1.** Distribution of average values (±SD) in balance tests for underweight, differentiated by gender.

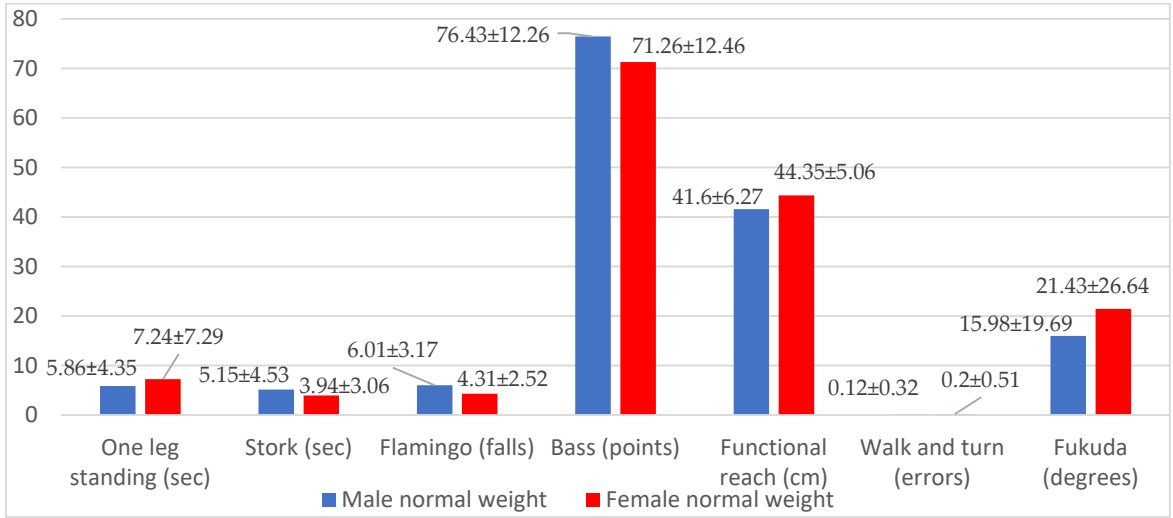

**Figure 2.** Distribution of average values (±SD) in the balance tests for normal weight students, differentiated by gender.

At the level of the normal weight group, the superiority of women in the one leg standing test and of men in the stork test is also found, but the average differences between genders are lower in value. Boys confirm better results for the bass test, Fukuda test and walk and turn field sobriety test, and women have superior performance in the flamingo test and functional reach test.

A different situation is presented for overweight male and female case (Figure 3).

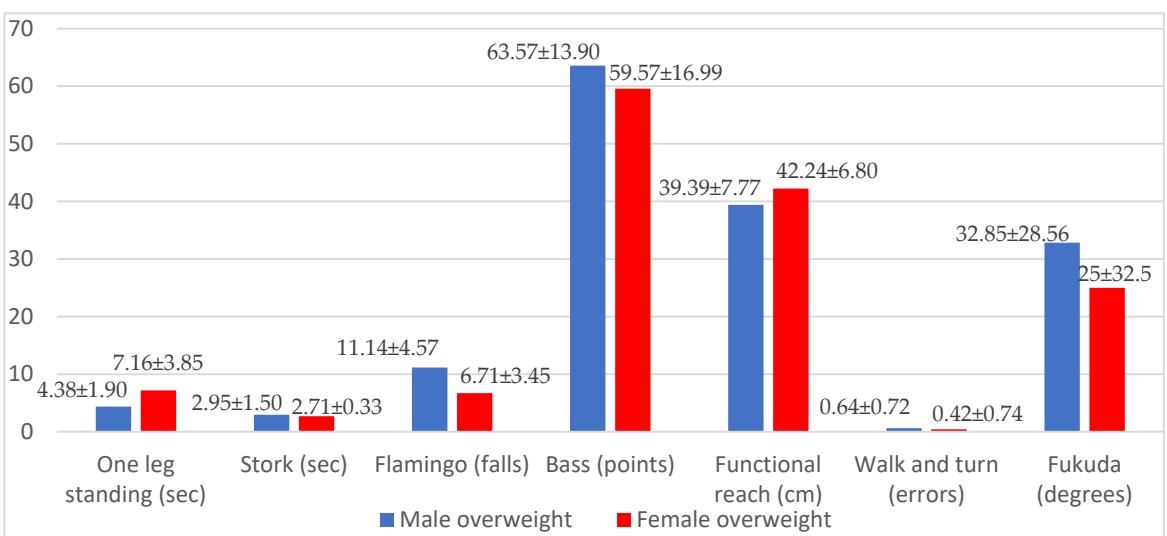

**Figure 3.** Distribution of average values (±SD) in balance tests for overweight, differentiated by gender.

For overweight students, the same distribution of gender superiority is observed for the 3 static balance assessment tests, with greater difficulties related to the flamingo test for men, as well as their poorer performance in the functional reach test, walk and turn field sobriety test and Fukuda test. Therefore, most overweight women score better on most balance assessment tests than overweight men, according to Figure 3. By comparison, Figures 1 and 2 show a relative balance in the distribution of performances in gender, underweight and normal weight tests.

## 4. Discussion

The data provided by the studied scientific papers offer different age category, gender and BMI level results largely similar to those of our study, but with obvious peculiarities/variations in the evolution of balance depending on the particularities of the investigated groups and the level of physical activism.

Increased BMI values and the inclusion of children (12–15 years old) in overweight and obese classes will lead to reduced balance ability in the anterior reach direction compared to normal weight, as well as differences in muscle strength of the flexors and extensors of the lower limbs [65]. There is a negative effect of excess weight on postural control and decreased efficiency of upper limb movements. Improving the muscular strength of the lower limbs for obese people does not solve the problem, as indicated by weight loss, for a better control of the balance and efficient movements of the upper limbs [66]. Other research finds weaker values of static and dynamic balance skills for overweight boys (8–10 years) compared to their normal weight counterparts [67]. Our research identifies the association of high BMI values with poor results in most balance tests, with the exception of the one leg standing test and the functional reach test.

The problems of obese people, whose balance is more pronounced than those of normal weight are reported by numerous studies. Significant weight loss, even if this reduces the values of the isometric knee extension, generates an increase in performance in the manifestation of balance and postural control, which can be optimized by physical activities that generate decreased BMI [68]. Another study which targeted a group of young males (10–21 years old) identifies problems of maintaining balance for overweight subjects, via comparison with the values of those of normal weight, applying the Bruininks-Oseretsky set of tests [69]. Our study confirms balance problems of the overweight, for the studied age category, for most of the applied tests.

A study of teenagers in Kosovo on the results of Eurofit tests found that girls performed better on static balance (Flamingo test) only in the 14–15 age group; for the rest of

the age groups boys performed better, but the differences between the sexes are statistically insignificant [70]. In the flamingo test, another study identifies, for the 13–15 age group, 13.6 falls for Kosovo students and 12.6 falls for Montenegrin students, with balanced results by gender [71]. For pubescent children (10–12 years) the use of balance exercises for 8 weeks × 3 days/week improves performance on speed, agility and static balance (flamingo test). Girls have poorer results compared to boys in the initial tests, but the values are slightly better in the final ones [72]. However, we recorded for the flamingo test, the certain superiority of women over men and the very poor results of overweight.

In an analysis of the influence of gender and BMI levels on balance values for children in China (8–10 years), no significant differences were found between genders and BMI classes (normal weight and obese) to maintain static bipodal balance with the eyes closed, but for the unipodal one with the eyes open, there are significant differences between genders and BMI classes. Dynamic balance assessment generates significant differences between obese and normal weight, but not between genders [73]. However, we identified superior values of women in the one standing balance test (with eyes closed), and the values of normal weight and underweight are significantly better than those of overweight. For most dynamic balance assessment tests, our research also does not find significant differences between the genders, except for the functional reach test, which is also in favor of women.

An investigation carried out on children aged 6–12 indicates significant differences in dynamic balance performance on the Bass test, between obese people with or without a flat foot and underweight with a flat foot. However, no differences were found between the groups for assessing static balance using the stork balance stand test. Obese children with flat feet have effected both static and dynamic balance, and overweight children with flat feet are affected only in terms of static balance [74]. The efficiency of slackline training for boys (8–14 years), in terms of balance, highlights that the manifestation of effects of significant improvement of static balance (Stork test) on unstable surfaces (air cushion) are found, but the effects are weak or zero for dynamic balance (Bass test) [75]. Other research aims at investigating static balance reference values assessed using the stork test for Spanish children aged 3–6 years. Girls perform better than boys at 4 years old, but poorer at 6 years old, and aging increases/improves the performance of the whole group studied [76]. We obtained higher values of the stork test and Bass test for men, even if they are insignificant.

A high level of PA (physical activity) provides better fitness values for obese people with high blood pressure (HT) and over the age of 40, including for the one leg standing test [77]. Obese young adults have a high incidence of unwanted postural changes (hyper-kyphosis, hyperlordosis and head protrusion). Their results in the dynamic balance tests are weaker, so they have a high risk of falling, high average time to perform tasks and accentuated dynamic mid-lateral movement. However, no significant differences were found between obese and normal weight subjects for unipodal and bipodal static balance, according to [78]. However, we found significant differences for most tests that assess unipodal static balance, in favor of normal weight and underweight.

Anthropometric changes, body weight gain and limitation of ROM/range of motion causes postural deviations in obese people; they also have a lack of strength that could ensure postural control and limited resistance to muscle fatigue, and the incidence of falls increases with the increase of body mass [79]. Our research strengthens this idea, through the poor results of overweight participants in tests that require good values of isometric force, in order to keep the body in those positions (flamingo test, stork test and one standing balance test).

For obese people, the high percentage of fat and poorly developed muscles will affect walking, neuromuscular feedback and maintaining orthostatic posture, with major risks of falls and injuries. The movement of the CoP/center of pressure is more obvious in obese people in the AP/antero-posterior direction, a significantly higher postural instability being found compared to the normal weight and overweight, according to [80]. Our study

showed a significantly higher number of errors committed by those who are overweight in maintaining the balance of the body moving on a line (walk and turn field sobriety test), but we did not find significant differences between genders.

Physical activity improves balance. Significant differences are found between sedentary university students and those who play football, without major differences between genders being reported according to [81]. A study in young adults (x = 21 years) indicates that subjects who have moderate to vigorous physical activity have better balance values (the balance area is smaller). Women have better scores than men, and the open-eyed version outperforms the closed-eyed version. Reducing sedentary behavior can improve static balance [82].

The modified functional reach test can be applied to assess balance in people who have suffered a stroke at 2–8 weeks after this incident/post event [83]. The usefulness of the functional reach test as a viable tool for measuring balance in children (5–15 years) is supported by [84], although other sources also identify the role of spinal flexibility in test performance. Highlighting the poorer performance of obese people (both genders) for the functional reach test, as well as impaired walking quality, compared to normal weight subjects is supported by [85]. We did not find significant differences between the BMI levels in this test (even if the normal weight participants have the best results), but we do report significant differences between the genders, with superior performance of women.

Problems related to maintaining static balance in the standing one leg and standing on tiptoes variants for children in Brazil (6–9 years old), but also the determination of some negative correlations between high BMI/overweight values and values on balance tests, are mentioned by [86]. The investigation, performed on young athletes (x = 21.34 years), shows positive correlations between BMI, body mass and height with dynamic balance. Even if athletes with high BMI and body mass values achieve better postural balance, they have greater problems when they lose their balance and try to rebalance their body. The authors recommend for those of tall stature to be involved in several balance workouts, for success in dynamic sports activities [87]. We also found lower balance values for those with high BMI values, in the applied dynamic tests.

The chances of a child having balance problems will increase with BMI values, so controlling body weight in children (6–12 years) through physical activity is very important. The link between high BMI values and the low fitness level of low-income children in Ghana and South Africa—as a predictor of limited motor skills—is highlighted by [88]. An analysis of the relationship between postural stability with BMI and gender variables for healthy young people (x = 21.7 years) found that women have greater postural sway than men. Obese people perform poorly on bipodal and unipodal balance tests compared to underweight and normal weight people [89]. This aspect was also confirmed by our study.

A study aimed at gender differences in the expression of motor skills of elementary school students (6–8 years) highlighted that girls perform better in static and dynamic balance tests and boys in the agility tests [90]. Our research indicates the superior results of boys in the Bass test (which involves alternating jumps and changing direction, from one foot to the other, while maintaining balance), so it confirms a better agility, but without statistical significance.

The usefulness of the Fukuda test as a predictor of benign paroxysmal postural vertigo is highlighted by [91]. Although the test has a lower sensitivity, it is useful in signaling/diagnosing mild and moderate unilateral vestibular dysfunctions. Our study identifies such issues only for overweight people, especially men.

The balance is better with age; a study on Turkish athletes (x = 13 years old) showed better scores for those who do individual sports at the expense of sports games, but where there are better reaction times [92]. Investigations about gender and age stage balance highlight that there are similar values recorded only for the 20–49 years age range, with decreases after 50 years. Men perform slightly better than women, and normal weight women have better scores than overweight women [93]. We identified similar results, but with insignificant superiority of men only for 3 situations (stork test, Bass test and Fukuda test).

Judo practitioners and sports dancers perform better on open-eye balance tests compared to non-sports groups, which demonstrates the importance of sports activities in optimizing balance. However, judokas have higher values than dancers in closed-eye tests, as an effect of proprioceptive adaptations, with implications in maintaining balance [94]. Gymnasts have the best balance performance, followed by football players, swimmers and basketball players. Top golf, archery or rifle athletes have superior balancing skills than their peers in the lower echelon of performance. Balance training positively influences the performance of vertical jumps and agility tests, at the level of physical education students and for those who are involved in practical recreational activities [95]. Similar values of static and dynamic balance between gymnasts and footballers (soccer) are found at the level of college students, while for basketball players there are weaker values of static balance compared to gymnasts and of dynamic balance compared to footballers [96]. The comparison of balance values (in 2004 and 2014) at the flamingo test, for Polish university students in physical education and sports, showed an insignificant decrease in performance for girls, but significant for men. For both groups studied, girls have better values than boys in this test [97], which is similar to our results.

## 5. Conclusions

The results of our investigation are consistent with and reinforce the ideas of other similar studies, which highlight the negative impact of high BMI values on the quality of movements and static and dynamic postural stability for different categories of the population studied.

The analysis of variance and the comparison of the average values at the level of the gender variable indicate a balance in the distribution of the results, without one of the sexes having higher scores for all the tests. However, women are the only ones with significantly better results for the one leg standing test, functional reach test and flamingo test (with F values associated with thresholds $p < 0.05$). Even if men get slightly higher average scores on the stork test, Bass test and Fukuda test, these differences are not statistically significant (F values are associated with thresholds $p > 0.05$).

Comparisons between pairs at the BMI step variable confirm the problems that the overweight group has in maintaining balance. For almost all tests (except the functional reach test) significant values of F and significant differences between the mean values in tests between normal weight and overweight or between underweight and overweight participants ($p < 0.05$) are reported. Overweight people have the greatest difficulty maintaining static balance and make the most mistakes in dynamic balance tests. However, a surprise of the study is the very good scores of the underweight subjects in the one leg standing test, flamingo test and walk and turn field sobriety test, where their values are superior to normal weight students, and in the other tests they obtain results close to theirs.

The analysis of the correlations between BMI values and performance in balance tests indicates negative associations between these variables for most cases, so the increase in BMI is correlated with poorer performance in assessing balance and an increase in errors found in the set of tests. Our study confirms the obvious difficulties of taking balance tests for overweight and obese subjects, aspects that negatively influence the safe and efficient execution of movements in daily activities and sports, for this category of subjects that unfortunately is more and more present nowadays.

### *Limits of the Study and Future Research Directions*

The low percentage of underweight students is the main limitation of this study, which does not allow us to generalize the results obtained, and a repetition of the research on representative groups in this category could bring additional information in this regard. Most of the consulted studies focus on the balance issues of obese and overweight participants or on comparisons between different branches of sport, but the category of underweight is neglected (probably as a result of poor representation in the population). Another sensitive point of our study is the lack of laboratory tests (using technologies based on sensors and

baropodometric platforms) and which would have facilitated the collection of data with high accuracy, regarding: CoP position, its oscillations in the antero-posterior and middle-lateral plane, the control of the muscle groups involved primarily in static and dynamic effort, the pressure exerted on the joints and the distribution of body weight on the plantar areas, etc. In this regard, recent studies that offer new directions for investigating body balance are being conducted by [98–100]. Testing by using balls of different shapes and inflated at different pressures to develop balance is another possible direction, according to the studies of [101,102].

**Author Contributions:** Conceptualization: G.D.M. and G.M.; methodology: G.D.M.; software: G.D.M. and G.M.; formal analysis: G.D.M.; investigation G.D.M.; resources: G.D.M.; data curation G.D.M.; writing—original draft preparation, G.D.M.; writing—review and editing G.D.M. and G.M.; visualization, G.D.M. and G.M.; supervision G.D.M.; project administration, G.D.M. and G.M.; funding acquisition G.M. All authors have read and agreed to the published version of the manuscript.

**Funding:** This research received no external funding.

**Institutional Review Board Statement:** An institutional approval for studies involving humans was accessible. The approval of the Ethics Commission is registered according to No. 119/CEU/24.02.2022.

**Informed Consent Statement:** Consent was obtained from all subjects involved in the study and as a result we received the approval of the commission according to the request RCF 867/21.02.2022.

**Acknowledgments:** The authors of the article thank the students who participated in the balance assessment study.

**Conflicts of Interest:** The authors declare no conflict of interest.

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
