# Peer review of "The Association of Gender and Body Mass Index on the Values of Static and Dynamic Balance of University Students (A Cross-Sectional Design Study)"

_applsci, doi:10.3390/app12083770_

Round 1

Reviewer 1 Report

I have read with interest this paper where the Authors report data regarding the impact of gender and BMI on static and dynamic the balance in university students.

For such a reason the article should be suitable for publication, nevertheless some minor revisions are needed.

  1. Abstract

Please report more clearly which is the aim of the study, it is important underline this also in the abstract.

  1. Introduction

In my opinion is not correct to report the reference number without ending the phrase, this could lead to confusion in the readership. Please correct ending in the right way the sentences and put the references as appropriate.

  1. Figures (1,2,3) the Authors reported average values (istogram red and blue) it would be appropriate to ad also standard deviation.

Author Response

Dear reviewer,

First and foremost, thank you for your patience in reading our study and for your helpful formulated recommendations. We hereby attach the answers to your suggestions for improving the article, noting that the lines of the article and the references do not longer have the same number / indicator, due to changes made in the text. All the changes/modifications that you requested as well as the other 3 reviewers are in blue, in the new version of the article.

Comments and Suggestions for Authors

I have read with interest this paper where the Authors report data regarding the impact of gender and BMI on static and dynamic the balance in university students.

For such a reason the article should be suitable for publication, nevertheless some minor revisions are needed.

  1. Abstract

Please report more clearly which is the aim of the study, it is important underline this also in the abstract.

A: Thanks for the suggestion. The purpose of the research was mentioned in the abstract.

  1. Introduction

In my opinion is not correct to report the reference number without ending the phrase, this could lead to confusion in the readership. Please correct ending in the right way the sentences and put the references as appropriate.

A: Thank you for the indication. All the references (from introduction and discussion) have been changed, they now appear at the end of the respective sentences.

  1. Figures (1,2,3) the Authors reported average values (istogram red and blue) it would be appropriate to ad also standard deviation.

A: Thanks for the suggestion. The mentioned charts have been modified (SD / standard deviation values have also been added).

 Thank you for all the guidance and suggestions provided.

  Best regards,

  Professor Murariu Gabriel.

  Date:  30.03.2022

Reviewer 2 Report

The Article “The influence of gender and Body Mass Index on the values of static and dynamic balance of university students” requires several changes before it will be published. There are some remarks concerning this article:

  1. In whole article used not appropriate citation (e.g., line 32, 36, 55, 73, 90 etc.).
  2. The language has to be corrected, even formulation of the aim is presented not very clearly.
  3. I wonder if really all the students at age 18-22 years are having physical education at University’s or High schools?
  4. The estimation of BMI and dividing persons to underweight, normal weight and overweight/obesity groups not explained. The term used for different “weight classes/ weight stages” (line 128) is also not appropriate.
  5. The term of “balance” it is much more appropriate than “equilibrium” in this context (line 66, 139, 192).
  6. In table 3 presented influence of the gender variable on the performance in the balance data, but from the data it in not possible to understand in which test male or male presenting better results. In addition, some data from this table are repeated in table 5 (same is with 4 and 6 tables). Additionally, there is repetition with data on figures with not clear units presented on y axis (Figure 1, 2, 3).

Before publication, in my opinion, article must be improved.

Author Response

Dear reviewer,

First and foremost, thank you for your patience in reading our study and for your helpful formulated recommendations. We hereby attach the answers to your suggestions for improving the article, noting that the lines of the article and the references do not longer have the same number / indicator, due to changes made in the text. All the changes/modifications that you requested as well as the other 3 reviewers are in blue, in the new version of the article.

Comments and Suggestions for Authors

The Article “The influence of gender and Body Mass Index on the values of static and dynamic balance of university students” requires several changes before it will be published. There are some remarks concerning this article:

  1. In whole article used not appropriate citation (e.g., line 32, 36, 55, 73, 90 etc.).

A: Thank you for your comment. All the references (from introduction and discussions) have been changed; they now appear at the end of the respective sentences.

  1. The language has to be corrected, even formulation of the aim is presented not very clearly.

A: Thank you for the suggestion. After the completion of the changes requested by all the reviewers, the article will be submitted for final language correction to another licensed translator. If problems arise again from this point of view, please indicate the paragraphs / lines where you have identified their failure to resolve.

  1. I wonder if really all the students at age 18-22 years are having physical education at University’s or High schools?

A: Thanks for the question. Physical education is a compulsory subject by law in Romania, including in university education (LAW no. 69 from April 28th, 2000 / Law on physical education and sports). Article 5 of the mentioned law stipulates this aspect. All undergraduate (not master's) students in undergraduate education do physical education in the first 2 years of study. The Faculty of Physical Education and Sports ensures the teaching of this discipline for all the other faculties and specializations from the Lower Danube University of Galați. We attach the link (the information is in Romanian):

http://www.cdep.ro/pls/legis/legis_pck.htp_act_text?idt=22911

  1. The estimation of BMI and dividing persons to underweight, normal weight and overweight/obesity groups not explained. The term used for different “weight classes/ weight stages” (line 128) is also not appropriate.

A: Thanks for the suggestion. We mentioned the BMI calculation formula (weight (kg) / [height (m)]2)  in section 2.1 / Participants. The BMI values associated with each BMI step are listed in Table 1:

- Underweight - BMI (below 18.5)

- Normal weight - BMI (18.5-24.9)

- Overweight - BMI (25-30)

- Obesity - BMI (above30)

The mentioned terms (weight classes / weight stages) have been replaced by the term BMI levels, so the purpose of the research is now more clearly formulated.

  1. The term of “balance” it is much more appropriate than “equilibrium” in this context (line 66, 139, 192).

A: Thank you for the recommendation. I replaced it with the term "balance" in all the lines you mentioned (but they now have a different numbering).

  1. In table 3 presented influence of the gender variable on the performance in the balance data, but from the data it in not possible to understand in which test male or male presenting better results. In addition, some data from this table are repeated in table 5 (same is with 4 and 6 tables). Additionally, there is repetition with data on figures with not clear units presented on y axis (Figure 1, 2, 3).

A: Thank you for your comment. Tables 3 and 4 present the results / influence of each independent variable (gender, respectively BMI steps) upon the values of each test. It is normal for the results of Sum of Squares and Mean Square to be identical in Table 3, because the Gender variable is dichotomous / 2 steps (male and female). This also explains why the significance thresholds of F in Table 3 coincide with those of the pair differences in Table 5.

 (for example, in the Flamingo test, the value of F is associated with a threshold P = 0.000, identical to the significance threshold between a-b for the same test). For the BMI step variable that is trichotomous (underweight, normal weight and overweight / obese) the results cannot be repeated: in Table 4, Sum of Squares and Mean Square are different, and the significance thresholds associated with F values ​​in Table 4 are not necessarily identical to the 3 associated thresholds associated with the differences between the data pairs in Table 6 (a-b, a-c and b-c). Example: In Table 4, the general influence of BMI steps upon the Stork test values ​​indicates a value of F associated with a threshold P = 0.008, so statistically significant. However, the detailed analysis of the differences on the 3 data pairs in Table 6 is statistically insignificant for the first 2 data pairs (a-b / P = 1, respectively a-c / P = 0.326), but significant for the last compared data pair (b-c / P = 0.006), so only between normal weight and overweight / obese significant differences are found in this test. In summary, Tables 3 and 4 show test univariates (therefore a general influence of each variable: gender and BMI steps upon each separate balance test) and Tables 5 and 6 make a finer comparison between the steps of each independent variable (male-female, respectively underweight-normal-weight, underweight-overweight and normal-weight-overweight).

Figures 1, 2 and 3 present the data resulting from the interaction / combination of the 2 independent variables, which differ radically from those from the tables. We preferred the presentation for each BMI level of gender differences (i.e or meaning underweight women-underweight men, normal-weight women-normal-weight men, respectively overweight women-overweight men). The SPSS software presents the average and combination values, for example the differences that appear within the BMI steps for each gender (underweight men-normal-weight men-overweight men, or underweight women-normal-weight women-overweight women), but these data would have loaded excessively the volume of results presented in the study. The units of measurement for the tests were presented in the methodology; they are varied (sec, cm, degrees, drops, errors) and were mentioned next to each test in the mentioned graphs.

Before publication, in my opinion, article must be improved.

Thank you for all the guidance provided.

 Best regards,

  Professor Murariu Gabriel.

  Date  30.03.2022

Reviewer 3 Report

The topic seems interesting, so thank you for a chance of reviewing this study. 

Title is well-defined and matches the contents. 

Abstract is informative and key words are also ok. 

Introduction is generally well-written but in my opinion it would be good for your work if you expanded and mention some more alleys for a reader to broaden the context but also to strengthen the rationale for your study. 

In the introduction you lead the reader through different factors linked to the main topic, but there some are issues worth adding. For example, the quality of students entering university has change in last decades both in sense of their somatype (for example look into: Changes in the somatic build and physical fitness of physical education students in the years 2004-2014) or/and in terms of their levels of activity (for example: Physical activity rates of male and female students from selected European physical education universities, which has some impact on the factors that you have been measuring in your study (look into:  Comparison of the static balance, strength, and flexibility characteristics of the University Students who taken artistic gymnastic lesson and another paper: Comparison of Static and Dynamic Balance in Female Collegiate Soccer, Basketball, and Gymnastics Athletes. The observed decline in physical activity levels is connected with less fit young people who train sports (also coordination sports, balance sports) and in fact is becomes harder to make training progress with them. I know that you explore this path about effects of training on balance but this can be extended a bit, in my opinion linked with increasead sedentary bahaviours in youth, especially during last 2-3 years of pandemic. 

Methods - this section also requires some clarification and amendments. For example please provide how the sample group was selected in detail? You mention that the selection was randomized, but how? This would strengthen the value of findings and it weave the problems with imbalance between the BMI groups. 

In procedure please clarify the exact time during the day when the testing took place as the daytime may have an effect on stability, especially on the vitality of core muscles responsible for stability (for example: Assessing gait stability: The influence of state space reconstruction on inter- and intra-day reliability of local dynamic stability during over-ground walking). 

Statistical methods have been used correctly and its description is fine.

Results are well presented and clearly described. 

Discussion -  good analysis of own findings against those of other authors, especially exploring the links with obesity and balance problems. But again, the increased prevalence of obesity arises from some broader changes in human behaviours, specifically in terms of PA (pandemic, lack of support from PE teachers, parents ect. for example - how is classmate and PE teacher support associated with the level of physical activity in young adolescents from Kosovo? The role of gender and age? or PE teacher and classmate support in level of physical activity: The role of sex and BMI status in adolescents from Kosovo. And these issues are connected with youth spending less and less time outdoor for their leisure time PA. And as a consequence it deprives them from the daily contact with natural environment with increasing impact from the use of media (also on the hearing ability, special orientation, balance - see for example: Colorado's millennial generation: Youth perceptions and experiences of nature. Perhaps it would also be valuable input if you looked at the issues with balance stability in disabled people (see for example: (Micro) mobility, disability and resilience: Exploring well-being among youth with physical disabilities. 

These above mentioned links could be used to explore a bit broader spectrum of problems related with decreased PA of today's youth and perhaps to  replace those references mentioning studies on 6-8 years old or 13 years pupils (at the end of the Discussion section), unless, you believe they are essential for the line of your argumentation. 

The limitations of your study provided at the end of the text reflect authors' good understanding of the weaknesses in their own research, which is always a valid point that shows the critical methodological thinking. 

Tables are neat and easy to read.

I think with the enhancement of some lines of arguments and strengthening of the rationale. 

Author Response

Open Review

English language and style

( ) Extensive editing of English language and style required
( ) Moderate English changes required
(x) English language and style are fine/minor spell check required
( ) I don't feel qualified to judge about the English language and style

Yes

Can be improved

Must be improved

Not applicable

Does the introduction provide sufficient background and include all relevant references?

( )

( )

(x)

( )

Is the research design appropriate?

( )

(x)

( )

( )

Are the methods adequately described?

( )

(x)

( )

( )

Are the results clearly presented?

(x)

( )

( )

( )

Are the conclusions supported by the results?

( )

(x)

( )

( )

Dear reviewer,

First and foremost, thank you for your patience in reading our study and for your helpful formulated recommendations. We hereby attach the answers to your suggestions for improving the article, noting that the lines of the article and the references do not longer have the same number / indicator, due to changes made in the text. All the changes/modifications that you requested as well as the other 3 reviewers are in blue, in the new version of the article.

Comments and Suggestions for Authors

The topic seems interesting, so thank you for a chance of reviewing this study. 

Title is well-defined and matches the contents. 

A: Thank you for your appreciation.

Abstract is informative and key words are also ok. 

A: Thank you for your appreciation.

Introduction is generally well-written but in my opinion it would be good for your work if you expanded and mention some more alleys for a reader to broaden the context but also to strengthen the rationale for your study. 

In the introduction you lead the reader through different factors linked to the main topic, but there some are issues worth adding. For example, the quality of students entering university has change in last decades both in sense of their somatype (for example look into: Changes in the somatic build and physical fitness of physical education students in the years 2004-2014) or/and in terms of their levels of activity (for example: Physical activity rates of male and female students from selected European physical education universities, which has some impact on the factors that you have been measuring in your study (look into:  Comparison of the static balance, strength, and flexibility characteristics of the University Students who taken artistic gymnastic lesson and another paper: Comparison of Static and Dynamic Balance in Female Collegiate Soccer, Basketball, and Gymnastics Athletes. The observed decline in physical activity levels is connected with less fit young people who train sports (also coordination sports, balance sports) and in fact is becomes harder to make training progress with them. I know that you explore this path about effects of training on balance but this can be extended a bit, in my opinion linked with increased sedentary behaviours in youth, especially during last 2-3 years of pandemic. 

A: Thank you for your offered suggestions. The introduction has been extended according to your instructions ((however, some of the recommended titles for discussion were found in a more appropriate place here, and some of those indicated for introduction were better suited in the context of the Discussion section), being included in this part of the paper with the following new titles:

- Barton, K.S. Colorado’s Millennial Generation: Youth Perceptions and Experiences of Nature. Journal of Geography 2012, 111, 213–223, doi:10.1080/00221341.2011.652648.

- Bronikowski, M.; LaudaÅ„ska-KrzemiÅ„ska, I.; Bronikowska, M.; Morina, B. How Is Classmate and PE Teacher Support Associated with the Level of Physical Activity in Young Adolescents from Kosovo? The Role of Gender and Age. Central European Journal of Public Health 2015, 23, 252–257, doi:10.21101/cejph.a4174.

- Bronikowski, M.; Bronikowska, M.; Laudańska-Krzemińska, I.; Kantanista, A.; Morina, B.; Vehapi, S. PE Teacher and Classmate Support in Level of Physical Activity: The Role of Sex and BMI Status in Adolescents from Kosovo. BioMed Research International 2015, 2015, e290349, doi:10.1155/2015/290349.

- Bronikowski, M.; Gonzales-Gross, M.; Kleiner, K.; Knisel, E.; Martinkova, I.; Stache, A.; Kantanista, A.; Lòpez, D.C.; Konlechner, A. Physical Activity, Obesity and Health Programs in Selected European Countries. Studies in Physical Culture and Tourism 2008, 15, 9–17.

- Maciaszek, J.; Honsová, Š.; Knisel, E.; Epping, R.; OÅ‚piÅ„ska-Lischka, M.; MichaÅ‚, B.; Pospieszna, B. Physical Activity Rates of Male and Female Students from Selected European Physical Education Universities. Trends in Sport Sciences 2020 Vol.27 No.2 2020, 63–69, doi:10.23829/TSS.2020.27.2-3.

- Cancela Carral, J.M.; Ayán, C.; Sturzinger, L.; Gonzalez, G. Relationships Between Body Mass Index and Static and Dynamic Balance in Active and Inactive Older Adults. Journal of Geriatric Physical Therapy 2019, 42, E85, doi:10.1519/JPT.0000000000000195.

- Lipowicz, A.; Bugdol, M.N.; Szurmik, T.; Bibrowicz, K.; Kurzeja, P.; Mitas, A.W. Body Balance Analysis of Children and Youth with Intellectual Disabilities. Journal of Intellectual Disability Research 2019, 63, 1312–1323, doi:10.1111/jir.12671.

- Porcelli, P.; Ungar, M.; Liebenberg, L.; Trépanier, N. (Micro)Mobility, Disability and Resilience: Exploring Well-Being among Youth with Physical Disabilities. Disability & Society 2014, 29, 863–876, doi:10.1080/09687599.2014.902360.

- Han, J.; Anson, J.; Waddington, G.; Adams, R.; Liu, Y. The Role of Ankle Proprioception for Balance Control in Relation to Sports Performance and Injury. BioMed Research International 2015, 2015, e842804, doi:10.1155/2015/842804.

- Rojhani-Shirazi, Z.; Azadeh Mansoriyan, S.; Hosseini, S.V. The Effect of Balance Training on Clinical Balance Performance in Obese Patients Aged 20–50 Years Old Undergoing Sleeve Gastrectomy. Eur Surg 2016, 48, 105–109, doi:10.1007/s10353-015-0379-8.

- Davoodeh, S.; Sheikh, M.; Houminiyan Sharifabadi, D.; Bagherzadeh, F. The Effect of Wii Fit Exergames on Static Balance and Motor Competence in Obese and Non-Obese College Women. Acta Gymnica 2020, 50, 61–67, doi:10.5507/ag.2020.008.

 We hope that by using these sources we will offer a better and more complex picture of the factors that condition and influence the manifestation of balance, the problems generated by the sedentary lifestyle, the variations in body balance by age categories, differences between genders, balance values at normal weight, overweight and sports subjects, variants of programs based on physical activities and their effect, measurement / evaluation tests using modern technologies, etc.

Methods - this section also requires some clarification and amendments. For example please provide how the sample group was selected in detail? You mention that the selection was randomized, but how? This would strengthen the value of findings and it weave the problems with imbalance between the BMI groups. 

A: Thank you for your guidance. For volume sampling n, the standard error (Std. Deviation) is the indicator that shows how accurately approximates the average calculated from the values ​​of a series, the average of the population from which the sample was extracted or the batch on which the measurements were made. An approximation of the average of a population is basically, the better the more numerous the extracted sample is. The volume of the considered group represents approximately 25% of the entire population of undergraduate students of the Faculties of Medicine and Computers - the largest faculties in the University. Under these conditions, due to the randomized selection by groups / specializations, respecting the proportion between genres and simple distribution, the inference obtained by using this sample is representative. We knew the numerical composition of the groups and the number of women / men in each group of the specializations stipulated in the paper, because they were sent by each faculty to the Faculty of Physical Education and Sports, in the order notes from the mouth of September. The other faculties from the University could not be included in the study, due to the fact that the research was carried out exclusively by the main author, who received physical education classes only at the 2 mentioned faculties, in the academic year 2018-2019.

In procedure please clarify the exact time during the day when the testing took place as the dayti me may have an effect on stability, especially on the vitality of core muscles responsible for stability (for example: Assessing gait stability: The influence of state space reconstruction on inter- and intra-day reliability of local dynamic stability during over-ground walking). 

A: Thanks for the suggestion. We know that circadian rhythm influences the physical performance and motor manifestation (not just for balance). Balance is also influenced by other variables (body temperature, fatigue, fitness level, age, etc.). The tests of the groups I worked with were scheduled between 12-16 PM, thus avoiding the evaluation very early in the morning and very late in the evening, when the performances would have been affected. The following sources were referred to the procedure in order to justify the chosen time slot:

- Amir, D.; Amir, H.S.; Saeed, G.; Amir, G.R. The Effect of Diurnal Rhythms on Static and - ProQuest. Biomedical Human Kinetic 2021, 13, 205–211, doi:0.2478/bhk-2021-0025.

- Karagul, O.; Nalcakan, G.R.; Dogru, Y.; Tas, M. EFFECTS OF CIRCADIAN RHYTHM ON BALANCE PERFORMANCE. Polish Journal of Sport and Tourism 2017, 24, 155–161, doi:10.1515/pjst-2017-0016.

 - Zouabi, A.; Quarck, G.; Martin, T.; Grespinet, M.; Gauthier, A. Is There a Circadian Rhythm of Postural Control and Perception of the Vertical? Chronobiology International 2016, 33, 1320–1330, doi:10.1080/07420528.2016.1215993.

- Cagno, A. di; Fiorilli, G.; Iuliano, E.; Aquino, G.; Giombini, A.; Battaglia, C.; Piazza, M.; Tsopani, D.; Calcagno, G. Time-of-Day Effects on Static and Dynamic Balance in Elite Junior Athletes and Untrained Adolescents. International Journal of Sports Science & Coaching 2014, 9, 615–625, doi:10.1260/1747-9541.9.4.615.

Statistical methods have been used correctly and its description is fine.

A: Thank you for your appreciation.

Results are well presented and clearly described. 

A: Thank you for your suggestions.

Discussion - good analysis of own findings against those of other authors, especially exploring the links with obesity and balance problems. But again, the increased prevalence of obesity arises from some broader changes in human behaviours, specifically in terms of PA (pandemic, lack of support from PE teachers, parents ect. for example - How is classmate and PE teacher support associated with the level of physical activity in young adolescents from Kosovo? / The role of gender and age? or PE teacher and classmate support in level of physical activity: The role of sex and BMI status in adolescents from Kosovo. And these issues are connected with youth spending less and less time outdoor for their leisure time PA. And as a consequence it deprives them from the daily contact with natural environment with increasing impact from the use of media (also on the hearing ability, special orientation, balance - see for example: Colorado's millennial generation: Youth perceptions and experiences of nature. Perhaps it would also be valuable input if you looked at the issues with balance stability in disabled people (see for example: (Micro) mobility, disability and resilience: Exploring well-being among youth with physical disabilities. 

These above mentioned links could be used to explore a bit broader spectrum of problems related with decreased PA of today's youth and perhaps to  replace those references mentioning studies on 6-8 years old or 13 years pupils (at the end of the Discussion section), unless, you believe they are essential for the line of your argumentation. 

A: Thanks for the suggestions that you made. The recommended titles were used in the introduction and / or discussion, to which we added other sources, for solid arguments and multiple comparisons with our results. The connection of the analysed sources in the Discussions, with the values ​​obtained by us in the balance tests is highlighted in green font. We did not remove any of the studies that are performed on other age groups, in order to notice similar or different aspects, but also because most research that addresses body balance is aimed at those with postural problems / various disabilities, overweight / obese or studying balance for performance athletes in various sports trials /sporting events. I found few studies for unsportsmanlike students in this age group. The last paragraph identifies research at similar ages, but on students in physical education and sports or athletes. In this part of the paper we have introduced the following research relevant to the approached topic or the subject :

  • Krzysztoszek, J.; Maciaszek, J.; Bronikowski, M.; Karasiewicz, M.; LaudaÅ„ska-KrzemiÅ„ska, I. Comparison of Fitness and Physical Activity Levels of Obese People with Hypertension. Applied Sciences 2021, 11, 10330, doi:10.3390/app112110330.
  • Perrin, P.; Deviterne, D.; Hugel, F.; Perrot, C. Judo, Better than Dance, Develops Sensorimotor Adaptabilities Involved in Balance Control. Gait & Posture 2002, 15, 187–194, doi:10.1016/S0966-6362(01)00149-7.
  • Hrysomallis, C. Balance Ability and Athletic Performance. Sports Med 2011, 41, 221–232, doi:10.2165/11538560-000000000-00000.
  • Bressel, E.; Yonker, J.C.; Kras, J.; Heath, E.M. Comparison of Static and Dynamic Balance in Female Collegiate Soccer, Basketball, and Gymnastics Athletes. J Athl Train 2007, 42, 42–46.
  • Wasiluk, A.; Saczuk, J. Changes in the Somatic Build and Physical Fitness of Physical Education Students in the Years 2004 and 2014. Biblioteka Akademii Wychowania Fizycznego w Poznaniu 2020.

The limitations of your study provided at the end of the text reflect authors' good understanding of the weaknesses in their own research, which is always a valid point that shows the critical methodological thinking. 

A: Thank you for your appreciation.

Tables are neat and easy to read.

A: Thank you for your appreciation.

I think with the enhancement of some lines of arguments and strengthening of the rationale.

 Thank you for all the guidance provided.

  Best regards,

  Professor Murariu Gabriel.

  Date 30.03.2022

Reviewer 4 Report

Since this is an observational design study, it is suggested that the title could replace the word "influence" with "association". In addition, it is recommended that the title could indicate that this is a cross-sectional design study.

It is suggested, in the "abstract", to clearly define the objective of the study. In addition, this section may begin with a brief contextualization of the problem under study.

The "Introduction" section has weaknesses regarding the contextualization of the object of study, being too extensive and sometimes redundant in its content. Furthermore, this report focuses on the relationships between BMI, gender and dynamic balance.  However, no studies related to the variables under study are presented.  As such, it would be preferable for the "Introduction" to begin with a contextualization about these variables. The report does not reference previously published studies in order to broaden the context and provide an empirical basis for subsequent hypothesis development. It is suggested that it could explain how the present study aims to overcome the methodological limitations of previous studies, thus contributing to the justification and relevance of the present study. The literature review presented is too extensive. It is suggested to synthesize this section in order to make the reading more comprehensible for the reader. It is recommended that its substantiation of the hypotheses be structured in such a way as to convince the reader of its logic and consistency with a different depth. Thus, it would be useful to describe or allude to the premises, concepts or theoretical models that relate to the object of study. In another dimension, it is suggested that the presentation of the relevance of the main subject under study to current practical knowledge could be further developed, supported by other evidence, previously published, giving it better robustness. The manuscript presents several formatting errors regarding the referencing system required by the journal: line 32" The static standing balance is a major component of motor fitness, and the investigations made by [2] ".

To better understand the sample universe, it will be important to clarify what selection and exclusion criteria were defined. Since sample size plays an important role in the ability to make accurate inferences, were any statistical procedures performed to calculate the sample size? The participants were selected randomly. However, what was the random sampling strategy used? It is strongly recommended that references of the validity and scientific reliability studies of the balance tests used (One leg standing with eyes closed/sec, Stork test/sec and Flamingo test/falls) be provided."

The presentation of results by textual form is not perceptible to the reader. Besides, the tables and graphs sometimes show redundant data duplicated in relation to the text. It is suggested to reformulate the presentation of the results.

The authors compared their results with studies carried out in child populations, and therefore, in very different motor development stages from the adult population assessed in this study. The "Discussion" section could present a more comprehensive development and relate the results of the present investigation to previously published studies related to the object of study, taking into consideration the ages of the participants. Furthermore, explanations based on theoretical models could be put forward for the verified results. The conclusion of the study is not perceptible. It is recommended that, instead of presenting a repetition of the results mentioned in the previous sections, a more compressive and succinct general interpretation of the results evidenced here be performed.

Author Response

pen Review

English language and style

( ) Extensive editing of English language and style required
( ) Moderate English changes required
( ) English language and style are fine/minor spell check required
(x) I don't feel qualified to judge about the English language and style

Yes

Can be improved

Must be improved

Not applicable

Does the introduction provide sufficient background and include all relevant references?

( )

( )

(x)

( )

Is the research design appropriate?

( )

(x)

( )

( )

Are the methods adequately described?

( )

( )

(x)

( )

Are the results clearly presented?

( )

( )

(x)

( )

Are the conclusions supported by the results?

( )

(x)

( )

( )

Dear reviewer,

First and foremost, thank you for your patience in reading our study and for your helpful formulated recommendations. We hereby attach the answers to your suggestions for improving the article, noting that the lines of the article and the references do not longer have the same number / indicator, due to changes made in the text. All the changes/modifications that you requested as well as the other 3 reviewers are in blue, in the new version of the article.

Comments and Suggestions for Authors

Since this is an observational design study, it is suggested that the title could replace the word "influence" with "association". In addition, it is recommended that the title could indicate that this is a cross-sectional design study.

A: Thank you for the recommendations. The title has been modified as indicated: The association of gender and Body Mass Index on the values of static and dynamic balance of university students (a cross-sectional design study)

It is suggested, in the "abstract", to clearly define the objective of the study. In addition, this section may begin with a brief contextualization of the problem under study.

A: Thanks for the suggestions. We have introduced in the abstract a phrase related to the importance of balance in the motor activities and the mention of the purpose of our study.

The "Introduction" section has weaknesses regarding the contextualization of the object of study, being too extensive and sometimes redundant in its content. Furthermore, this report focuses on the relationships between BMI, gender and dynamic balance.  However, no studies related to the variables under study are presented.  As such, it would be preferable for the "Introduction" to begin with a contextualization about these variables. The report does not reference previously published studies in order to broaden the context and provide an empirical basis for subsequent hypothesis development. It is suggested that it could explain how the present study aims to overcome the methodological limitations of previous studies, thus contributing to the justification and relevance of the present study. The literature review presented is too extensive. It is suggested to synthesize this section in order to make the reading more comprehensible for the reader. It is recommended that its substantiation of the hypotheses be structured in such a way as to convince the reader of its logic and consistency with a different depth. Thus, it would be useful to describe or allude to the premises, concepts or theoretical models that relate to the object of study. In another dimension, it is suggested that the presentation of the relevance of the main subject under study to current practical knowledge could be further developed, supported by other evidence, previously published, giving it better robustness. The manuscript presents several formatting errors regarding the referencing system required by the journal: line 32" The static standing balance is a major component of motor fitness, and the investigations made by [2] ".

A: Thanks for the suggestions. Even though there are many sources analysed, reviewer 3 considered that the introduction should be extended, and we are recommended to analyse and use certain sources. We have tried to group logically the sources in the introduction, focusing on various topics related to the manifestation of static and dynamic body balance: the role of balance in the performance of movements, the involvement of the 3 analysers / senses on which the balance depends, issues related to sedentary lifestyle and obesity in the manifestation of balance, some differences that appear between the sexes, the importance in sports activities and the comparison of the results between these studies, the evolution of balance at different age groups, the usefulness of certain training programs to optimize balance. In the introduction there were introduced the sources with the number: 5,6,7,10,11,13,21,24,26,39,43 (in the revised version of the manuscript). One argument that can support the usefulness of our study refers to the diversity of applied balance tests (7 tests), which allows the comparison of results with a multitude of other analysed research. All the reported formatting errors have been eliminated; the serial number of the cited authors is entered at the end of the sentences (in the introduction, methodology and discussions).

To better understand the sample universe, it will be important to clarify what selection and exclusion criteria were defined. Since sample size plays an important role in the ability to make accurate inferences, were any statistical procedures performed to calculate the sample size? The participants were selected randomly. However, what was the random sampling strategy used? It is strongly recommended that references of the validity and scientific reliability studies of the balance tests used (One leg standing with eyes closed/sec, Stork test/sec and Flamingo test/falls) be provided."

A: Thank you for the directions. For volume sampling n, the standard error (Std. Deviation) is the indicator that shows how accurately approximates the average calculated from the values ​​of a series, the average of the population from which the sample was extracted or the batch on which the measurements were made. An approximation of the average of a population is basically, the better the more numerous the extracted sample. The volume of the considered group represents approximately 25% of the entire population of undergraduate students of the Faculties of Medicine and Computers - the largest faculties in the University. Under these conditions, due to the randomized selection by groups / specializations, respecting the proportion between genres and simple distribution, the inference obtained by using this sample is representative. We knew the numerical composition of the groups and the number of women / men in each group of the specializations stipulated in the paper, because they are sent by each faculty to the Faculty of Physical Education and Sports, in the order notes from September. The other faculties from the University could not be included in the study, due to the fact that the research was carried out exclusively by the main author, who received physical education classes only at the 2 mentioned faculties, in the academic year 2018-2019.

We have studied and introduced a series of references of the studies of validity and scientific reliability of the 3 balance tests mentioned by you (in the procedure). These are the following:

- Curnow, D.; Cobbin, D.; Wyndham, J. Reliability of the Stork Test: Is Starting Stance Important? Chiropractic Journal of Australia 40, 137–141, doi:10.3316/informit.083750501229495.

- Muehlbauer, T.; Roth, R.; Mueller, S.; Granacher, U. Intra and Intersession Reliability of Balance Measures During One-Leg Standing in Young Adults. The Journal of Strength & Conditioning Research 2011, 25, 2228–2234, doi:10.1519/JSC.0b013e3181fb393b.

- Panta, K. A Study to Associate the Flamingo Test and the Stork Test in Measuring Static Balance on Healthy Adults. The Foot and Ankle Online Journal 2015, 5.

- Springer, B.A.; Marin, R.; Cyhan, T.; Roberts, H.; Gill, N.W. Normative Values for the Unipedal Stance Test with Eyes Open and Closed. J Geriatr Phys Ther 2007, 30, 8–15, doi:10.1519/00139143-200704000-00003.

- Tsigilis, N.; Douda, H.; Tokmakidis, S.P. Test-Retest Reliability of the Eurofit Test Battery Administered to University Students. Percept Mot Skills 2002, 95, 1295–1300, doi:10.2466/pms.2002.95.3f.1295.

The presentation of results by textual form is not perceptible to the reader. Besides, the tables and graphs sometimes show redundant data duplicated in relation to the text. It is suggested to reformulate the presentation of the results.

Thank you for your comment. Table 2 shows the multivariate test results. Tables 3 and 4 show the univariate test results / the influence of each independent variable (gender, respectively BMI steps) on the values ​​of each test. It is normal for the results of Sum of Squares and Mean Square to be identical in Table 3, because the Gender variable is dichotomous / 2 steps (male and female). This aspect also explains the reason for which the significance thresholds of F in Table 3 coincide with those of the differences in pairs in Table 5 (for example, in the Flamingo test, the value of F is associated with a threshold P = 0.000, identical to the significance threshold between a-b for the same test). For the BMI step variable that is trichotomous (underweight, normal weight and overweight / obese) the results are not repeated: in Table 4, Sum of Squares and Mean Square are different, and the significance thresholds associated with F values ​​in Table 4 are not required. identical to the 3 thresholds associated with the differences between the data pairs in Table 6 (a-b, a-c and b-c). Example: In Table 4, the general influence of BMI steps on Stork test values indicates a value of F associated with a threshold P = 0.008, so statistically significant. However, the detailed analysis of the differences on the 3 data pairs in Table 6 is statistically insignificant for the first 2 data pairs (a-b / P = 1, respectively ac / P = 0.326), but significant for the last data pair compared (b-c / P = 0.006), so only between normal weight and overweight / obese significant differences are found in this test. In summary, Tables 3 and 4 show test univariates (hence a general influence of each variable: gender and BMI steps on each separate equilibrium test) and Tables 5 and 6 make a finer comparison between the steps of each independent variable (male-female, respectively underweight-normal-weight, underweight-overweight and normal-weight-overweight). All tables had to be commented on, because a presentation of the relevant results had to be made, otherwise we do not think that their understanding would be easy.

Figures 1, 2 and 3 show the data resulting from the interaction / combination of the 2 independent variables, which differ radically from those in the tables. We preferred the presentation for each BMI level of gender differences (i.e underweight women-underweight men, normal-weight women-normal-weight men, respectively overweight women-overweight men). The SPSS software shows the average and combination values, for example the differences that occur within the BMI steps for each gender (underweight men-normal-weight men-overweight men, or underweight women-normal-weight women-overweight women), but these data would have loaded excessive volume of results presented in the study. The units of measurement for the tests were presented in the methodology; they are varied (sec, cm, degrees, drops, points, errors) and were passed next to each test in the mentioned graphs. SD values ​​on these charts were added at reviewer’s 1 request.

The authors compared their results with studies carried out in child populations, and therefore, in very different motor development stages from the adult population assessed in this study. The "Discussion" section could present a more comprehensive development and relate the results of the present investigation to previously published studies related to the object of study, taking into consideration the ages of the participants. Furthermore, explanations based on theoretical models could be put forward for the verified results.

A: Thank you for the suggestions. We have received a number of titles recommended by reviewer 3, which were used in the introduction and / or discussions, to which we added other sources, in order to have solid arguments and multiple comparisons with our results. The connection of the sources analyzed in the Discussions, with the values ​​obtained by us in the balance tests is highlighted in green font. We have tried to compare our results with those provided by the literature of specialty, even though the ages are often different. We have not removed anything of the studies that are performed on other age groups, in order to see similar or different aspects, but also because most research that addresses body balance is aimed at those with postural problems / various disabilities (being medical research), those overweight / obese, or study balance for performance athletes in various tests. I found few studies for unsportsmanlike students in this age group. The last paragraph identifies research at similar ages, but on students in physical education and sports or athletes (who certainly have superior motor skills).

The conclusion of the study is not perceptible. It is recommended that, instead of presenting a repetition of the results mentioned in the previous sections, a more compressive and succinct general interpretation of the results evidenced here be performed.

A: Thank you for the indication or suggestion. A short paragraph was introduced at the beginning of the conclusions, which summarizes the importance of our research. We would like to ask you that the other formulated ideas to remain please, being correlated with the 3 working hypotheses formulated (H1-influence of gender on equilibrium performance, H2-influence of BMI steps on performance, respectively H3-negative associations between BMI values and test results).

Thank you for all the guidance provided. If you would like to remove or add other sources, please let us know exactly which ones they are. The main author has searched numerous databases and believes that he has analyzed relevant research to the research topic, but any constructive recommendation can improve this material or paper. Can you talk to the other 3 reviewers in order to get a common point of view?

  Best regards,

  Professor Murariu Gabriel.

  Date 30.03.2022

Round 2

Reviewer 3 Report

Dear Authors 

I am glad you have done extensive changes to the text of your article and included most of the suggestions. I would suggest looking carefully at the references and adjust to the referencing style of the journal. In some references you give names and surnames and in others names and just initial of the surname and in other cases I am not sure wherther you give name of surname (like ref. 72, 55 - three times Amir? 36 Sebastia? 11 MichaÅ‚ ?    ). In same references titles are in Capital letters  (ref. 85 and 86) in others in small letters) ect. This needs to be checked again. 

Reviewer 4 Report

It turns out that the authors have made most of the changes requested in the previous review report. Congratulations.